# Characterization of Novel *Klebsiella* Phage PG14 and Its Antibiofilm Efficacy

Mansura S. Mulani,[a,b] Shital N. Kumkar,[a] Karishma R. Pardesi[a]

[a]Department of Microbiology, Savitribai Phule Pune University, Pune, Maharashtra, India
[b]Abeda Inamdar Senior College, Pune, Maharashtra, India

**ABSTRACT**　The increasing frequency of infections caused by multidrug-resistant *Klebsiella pneumoniae* demands the development of unconventional therapies. Here, we isolated, characterized, and sequenced a *Klebsiella* phage PG14 that infects and lyses carbapenem-resistant *K. pneumoniae* G14. Phage PG14 showed morphology similar to the phages belonging to the family *Siphoviridae*. The adsorption curve of phage PG14 showed more than 90% adsorption of phages on a host within 12 min. A latent period of 20 min and a burst size of 47 was observed in the one step growth curve. Phage PG14 is stable at a temperature below 30°C and in the pH range of 6 to 8. The PG14 genome showed no putative genes associated with virulence and antibiotic resistance. Additionally, it has shown lysis against 6 out of 13 isolates tested, suggesting the suitability of this phage for therapeutic applications. Phage PG14 showed more than a 7-log cycle reduction in *K. pneumoniae* planktonic cells after 24 h of treatment at a multiplicity of infection (MOI) of 10. The phage PG14 showed a significant inhibition and disruption of biofilm produced by *K. pneumoniae* G14. The promising results of this study nominate phage PG14 as a potential candidate for phage therapy.

**IMPORTANCE**　*Klebsiella pneumoniae* is one of the members of the ESKAPE (*Enterococcus faecium, Staphylococcus aureus, Klebsiella pneumoniae, Acinetobacter baumannii, Pseudomonas aeruginosa*, and *Enterobacter* species) group of pathogens and is responsible for nosocomial infections. The global increase of carbapenem-resistant *K. pneumoniae* has developed a substantial clinical threat because of the dearth of therapeutic choices available. *K. pneumoniae* is one of the commonly found bacteria responsible for biofilm-related infections. Due to the inherent tolerance of biofilms to antibiotics, there is a growing need to develop alternative strategies to control biofilm-associated infections. This study characterized a novel bacteriophage PG14, which can inhibit and disrupt the *K. pneumoniae* biofilm. The genome of phage PG14 does not show any putative genes related to antimicrobial resistance or virulence, making it a potential candidate for phage therapy. This study displays the possibility of treating infections caused by multidrug-resistant (MDR) isolates of *K. pneumoniae* using phage PG14 alone or combined with other therapeutic agents.

**KEYWORDS**　*Klebsiella* phage PG14, multidrug resistant, *Klebsiella pneumoniae*, phage therapy, biofilm inhibition and disruption

*K*lebsiella pneumoniae is one of the six difficult-to-treat pathogens included in the ESKAPE (*Enterococcus faecium, Staphylococcus aureus, Klebsiella pneumoniae, Acinetobacter baumannii, Pseudomonas aeruginosa*, and *Enterobacter* species) acronym (1). The emergence of carbapenem-resistant *K. pneumoniae* has demanded its inclusion in the World Health Organization's critical priority list of pathogens against which there is a dearth of effective antibiotics available. In addition to its alarming antibiotic-resistant nature, *K. pneumoniae* displays a high degree of virulence, enabling it to invade and survive in the host (2–4). The foremost virulence factor that ensures the survival of *K. pneumoniae* is the capsular polysaccharides (CPS). They play a crucial role in the evasion of the host defense and

Address correspondence to Karishma R. Pardesi, karishma@unipune.ac.in.

The authors declare no conflict of interest.

serve as a barrier in the form of a biofilm matrix that protects bacterial cells against antibiotic penetration (5).

Additionally, the adherence capacity of polysaccharide-rich biofilms generally leads to device-related persistent infections (6, 7). Apart from invasion assistance and antibiotic protection, biofilms provide nourishment and act as a medium for exchanging genetic material (8). The propensity to cause life-threatening infections with high mortality and morbidity in the era of scarcely available effective antibiotics highly accentuates the need to explore alternative approaches to control *K. pneumoniae* infections (9).

Before the discovery of antibiotics in the early 19th century, bacteriophages (or phages) were used to treat infectious diseases like dysentery. However, in the following decades, the discovery of antibiotics sidelined the importance of phage therapy. Nevertheless, the subsequent misuse of antibiotics introduced the world to the perils of antibiotic resistance, which compelled us to reconsider bacteriophages as an alternative to antibiotics (10, 11). The salient features worth mentioning that make phages convenient weapons against antibiotic-resistant pathogens include the ubiquitous occurrence of phages in all sorts of environments; the ability of phages to inhibit or disrupt the growth of planktonic cells and biofilms, respectively; and lastly, the need for a single dose to kill the targeted pathogen. Furthermore, numerous studies confirm the successful applications of individual phage preparation, phage cocktails, and bioengineered lysin proteins to cure topical and systemic bacterial infections (11–15).

There are several reports of phages that have been isolated and characterized against *K. pneumoniae* (16–19). Many studies have proved *in vitro* and *in vivo* efficacy of phages against *K. pneumoniae* (20–22). Case studies involving the effective use of *K. pneumoniae* phages as a therapeutic agent in combination with antibiotics in case of urinary tract infections (UTI), wound infections, and fracture-related infections are reported (23, 24). Most of the recently isolated phages against *K. pneumoniae* have shown expression of enzyme polysaccharide depolymerase, which plays an essential role in capsule degradation and thereby disrupt biofilms (16). Considering these potential applications of phages against biofilm-associated infections, we have isolated and characterized a phage named PG14 infecting *K. pneumoniae* and explored its antibiofilm activity. The future of phage therapy relies on developing a phage bank of characterized phages against multidrug resistant (MDR) bacterial pathogens. Characterizing phages is crucial to understand their action on bacterial hosts and their possible side effects on human and animal subjects if there are any (25). In this study, phage PG14 was characterized for its adsorption rate, latent period, burst size, host range, whole-genome sequence, and *in vitro* antibiofilm activity. The phage PG14 was able to inhibit and disrupt the biofilm of *K. pneumoniae* G14, making it a potential candidate for phage therapy. This study highlights the possibility of treating infections caused by MDR isolates of *K. pneumoniae* using phage PG14 alone or in combination with other therapeutic agents.

## RESULTS

**Isolation, purification, and identification of phage PG14 targeting *K. pneumoniae* G14.** Among the phages isolated against *K. pneumoniae*, *Klebsiella* phage PG14 was selected for further study based on its host range. A high-titer preparation of phage PG14 was prepared in SM buffer and stored for all further studies. Phage isolated against *K. pneumoniae* G14 showed clear circular plaques surrounded by a turbid halo zone (Fig. 1B inset). A transmission electron microscopic (TEM) image of the phage showed an icosahedral head (length, $82 \pm 5$ nm; width, $67 \pm 3$ nm) and tail (length, $133 \pm 10$ nm; width, $18 \pm 2$ nm) with tail fibers, therefore, suggesting that the phage belongs to the order *Caudovirales* and family *Siphoviridae* (Fig. 1B). The phage was named *Klebsiella* phage PG14 and has been abbreviated as phage PG14 in this article.

**Adsorption and one-step growth curve of phage PG14.** The scanning electron microscopic (SEM) image of phage PG14 adsorbed on *K. pneumoniae* G14 is shown in Fig. 1A. The adsorption curve of phage PG14 showed that more than 90% of phage PG14 (Fig. 1C) was adsorbed within 12 min. One-step growth curve of phage PG14 showed a latent period of 20 min and a burst size of 47 (Fig. 1D).

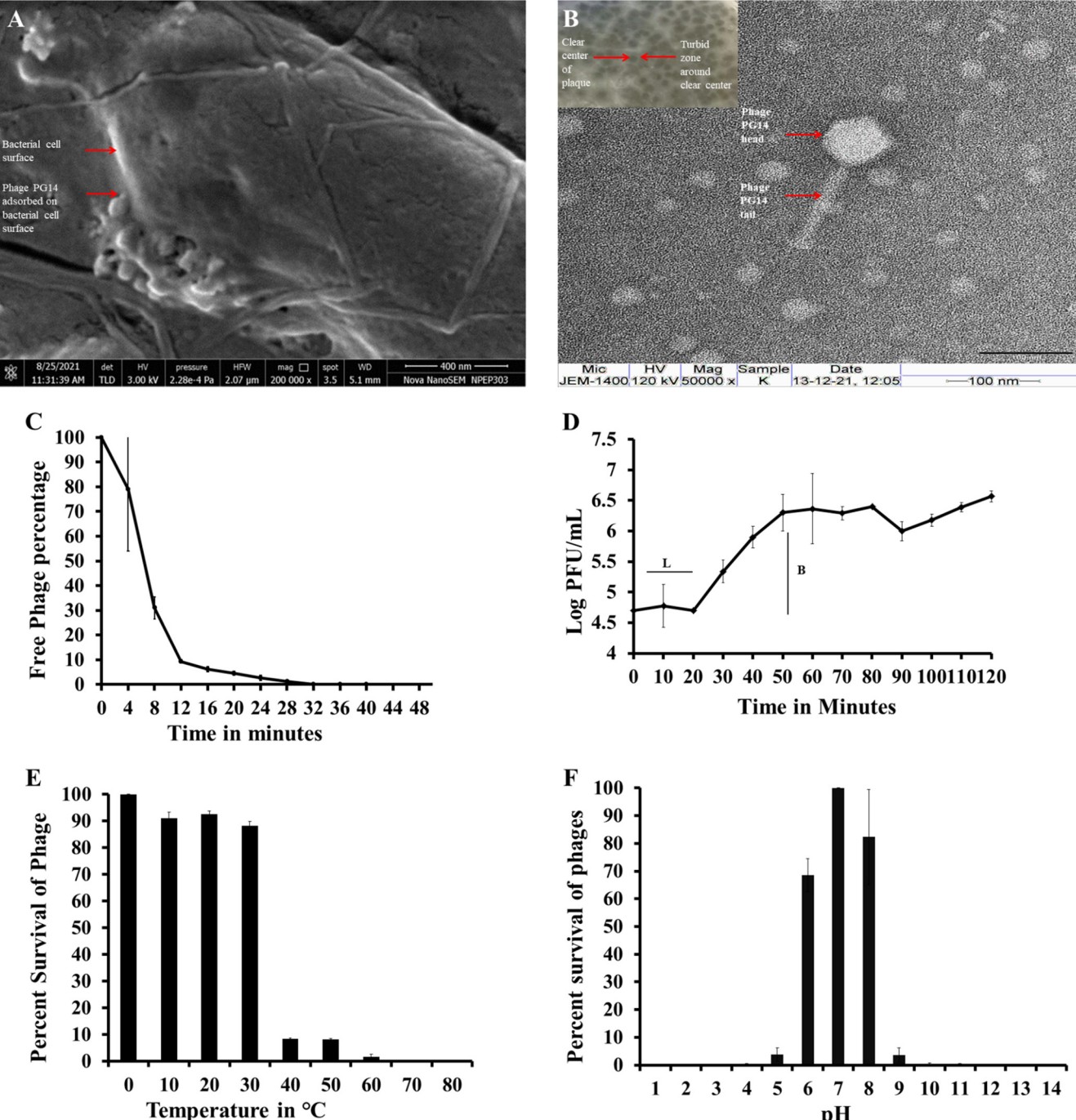

**FIG 1** Adsorption, morphology, one-step growth curve, and effect of temperature and pH on the viability of phage PG14. (A) Field emission scanning electron microscopic image at ×200,000 magnification showing phage PG14 particles adsorbed on the cell surface of *K. pneumoniae* G14. Red arrows show phage particle and bacterial cell surface, and the scale bar represents the scale of 400 nm. (B) Inset shows plaques of phage PG14, clear circular plaque surrounded by turbid halo zone; red arrows show a clear area at the center and surrounding turbid zone and transmission electron microscopic image of phage PG14 taken at ×50,000 magnification, showing an icosahedral head (length, 82 ± 5 nm; width, 67 ± 3 nm) and tail (length, 133 ± 10 nm; width, 18 ± 2 nm) with tail fibers; red arrows show phage head and tail, and scale bar represents the scale of 100 nm. (C) Adsorption curve of phage PG14, showing that more than 90% of phages got adsorbed within 12 min. L, latency period; B, burst size. (D) One-step growth of phage, showing the latent period of 20 min and burst size of 47. The influence of different temperature and pH range on phage PG14 viability when incubated for 60 min is shown in panels E and F, respectively. Phage PG14 is stable at a temperature below 30°C, and its viability is less than 10% above 40°C. Phage PG14 is stable in the pH range of 6 to 8 having more than 70% viability. In graph panels C, D, E, and F, data represent three independent experiments means ± standard deviations (SD).

**TABLE 1** Antibiotic resistance profile of host bacterial strains, host range, and efficiency of plating of phage PG14

| Name of bacterial strain[a] | Antibiotic resistance profile[b] | EOP |
|---|---|---|
| *K. pneumoniae* G14 (primary host) | AMP[r], AMC[r], TZP[r], CXM[r], CRO[r], SCF[i], FEP[r], IPM[i], MEM[r], AMK[s], GEN[s], NAL[r], CIP[r], CST[s], SXT[r] | 1 ± 0.75 |
| *K. pneumoniae* G5 | AMP[r], AMC[s], TZP[s], CXM[r], CRO[r], SCF[s], FEP[s], ETP[s], IPM[s], MEM[s], AMK[s], GEN[s], NAL[s], CIP[s], TGC[s], NIT[s], CST[s], SXT[s] | 0.8 ± 0.75 |
| *K. pneumoniae* S69 | AMP[r], AMC[r], TZP[r], CXM[r], CRO[r], SCF[r], FEP[r], ETP[r], IPM[i], MEM[r], AMK[r], GEN[r], NAL[r], CIP[r], TGC[s], NIT[r], CST[s], SXT[r] | 0.6 ± 0.75 |
| *K. pneumoniae* S71 | AMP[r], AMC[r], TZP[r], CXM[r], CRO[r], SCF[r], FEP[r], ETP[r], IPM[i], MEM[r], AMK[r], GEN[r], NAL[r], CIP[r], TGC[s], NIT[r], CST[s], SXT[r] | 0.7 ± 0.75 |
| *P. aeruginosa* G55 | TIM[i], TZP[s], CAZ[s], SCF[s], FEP[s], ATM[s], DOR[s], IPM[s], MEM[s], AMK[s], GEN[s], CIP[s], LVX[s], MIN[s], TGC[s], CST[s], SXT[s] | 0.1 ± 0.13 |
| *E. coli* S53 | AMP[r], AMC[s], TZP[s], CXM[r], CRO[r], SCF[s], FEP[r], ETP[s], IPM[s], MEM[s], AMK[s], GEN[s], NAL[r], CIP[r], TGC[s], NIT[s], CST[s], SXT[r] | 0.04 ± 0.04 |

[a]Isolates selected for determination of host range of phage PG14 were all members of the ESKAPE group of pathogens and were from in-house collection from the previous study. AMP, ampicillin; AMC, amoxicillin/clavulanic acid; TZP, piperacillin-tazobactam; TIM, ticarcillin/clavulanic; CAZ, ceftazidime; ATM, aztreonam; CXM, cefuroxime; CXM, cefuroxime axetil; CRO, ceftriaxone; SCF, cefoperazone/sulbactam; FEP, cefepime; ETP, ertapenem; IPM, imipenem; MEM, meropenem; DOR, doripenem; AMK, amikacin; GEN, gentamicin; NAL, nalidixic acid; CIP, ciprofloxacin; LVX, levofloxacin; MIN, minocycline; TGC, tigecycline; NIT, nitrofurantoin; CST, colistin; SXT, trimethoprim/ sulfamethoxazole.

[b]Resistant (r), intermediate (i), sensitive (s). All of the isolates enlisted in the table showed sensitivity to phage PG14 in the spot test, the EOP for these isolates were calculated as the ratio between PFU/mL on a sensitive strain and PFU/mL on the primary host (*K. pneumoniae* G14) and then ranked as "high productive" (≥0.5), "medium productive" (0.1 ≤ EOP < 0.5), "low productive" (0.001 < EOP < 0.1), and "inefficient" (<0.001) (43).

**Host range and efficiency of plating of phage PG14.** Phage PG14 was tested against 13 different bacterial isolates, *K. pneumoniae* ($n = 4$), *Staphylococcus aureus* ($n = 4$), *Acinetobacter baumannii* ($n = 2$), *Escherichia coli* ($n = 1$), and *Pseudomonas aeruginosa* ($n = 2$), of which, six including *K. pneumoniae* G14 showed sensitivity to phage PG14. Isolates selected for determining the host range of phage PG14 were members of the ESKAPE group of pathogens and were from in-house collections from the previous study. Antibiotic resistance profiles of six bacterial strains showing sensitivity to phage PG14 are given in Table 1, along with their efficiency of plating (EOP) with phage PG14.

**Stability of phage PG14 under abiotic stress conditions.** Temperature stability and pH studies demonstrated that the phage PG14 was stable at a temperature below 30°C (>85% viability), whereas its viability was reduced to less than 10% at temperatures above 40°C (Fig. 1E). The phage PG14 was stable between pH 6 and 8, showing >70% viability (Fig. 1F).

**Phage genome analysis.** The genome sequence of phage PG14 consists of 49,853 bp with a G+C content of 50.8%. The National Center for Biotechnology Information (NCBI) accession number for this sequence is OM964875. The genome contained 79 open reading frames, the majority of which are hypothetical proteins, and others include the phage structural proteins, enzymes, and proteins involved in DNA replication and repair, phage packing, and host lysis. The putative genes associated with lytic enzymes predicted in the phage PG14 genome included lysozyme, holin, and membrane-bound lytic murein transglycosylase, which have the potential to hydrolyze the bacterial cell wall. Positions of different proteins present in the phage PG14 are shown in the genome map given in Fig. 2. The proteomic tree, based on the whole-genome amino acid sequences of *Klebsiella* phage PG14 and closely related phages (Fig. 3A and 3B), showed that it is closely related to *Gammaproteobacteria* phages. Figure 3C indicates a heat map of OrthoANI values of phage PG14, with the other nine phages showing the highest similarity with *Klebsiella* phage NJS2.

**Time-kill assay for planktonic cells.** Planktonic cells of *K. pneumoniae* G14 were treated with phage PG14 with multiplicity of infection (MOI) values of 0.1, 1, and 10 at time intervals of 1, 3, 5, 7, and 24 h. A 0.4 to 7.8 log CFU/mL reduction was observed in treated samples compared to the control values. In addition, there was a significant difference between the treated and the control samples compared at each time interval and MOI ($P < 0.05$). Similarly, when log reduction at different MOIs was compared, a significant difference was observed only between 0.1 and 10 MOI when treated beyond 5 h ($P < 0.05$) (Fig. 4).

**Antibiofilm activity of phage PG14. (i) The biochemical characterization of the *K. pneumoniae*.** The biofilm matrix of bacteria generally includes polysaccharides, protein, and DNA. The biochemical characterization of the *K. pneumoniae* G14 biofilm matrix was done to determine its major component. Maximum reduction in biofilm was observed when the preformed *K. pneumoniae* G14 biofilm was treated with sodium meta periodate (an inorganic salt used to disrupt the extracellular polysaccharides),

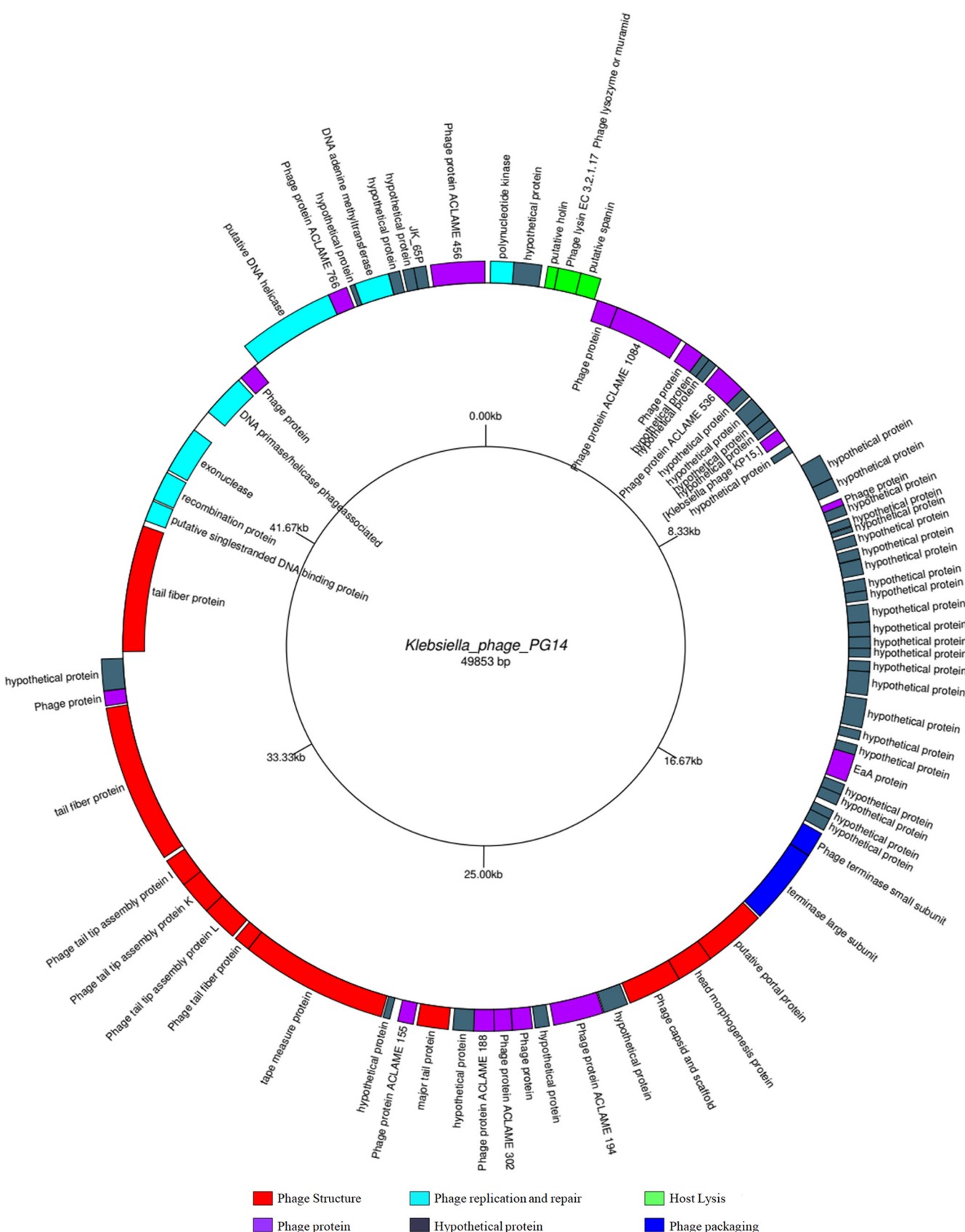

**FIG 2** Genomic map of *Klebsiella* phage PG14 created using tool GenomeVx; shows position and types of different proteins present in the genome. Red, phage structure; turquoise, phage replication and repair; light green, host lysis; purple, phage protein; gray, hypothetical protein; dark blue, phage packaging.

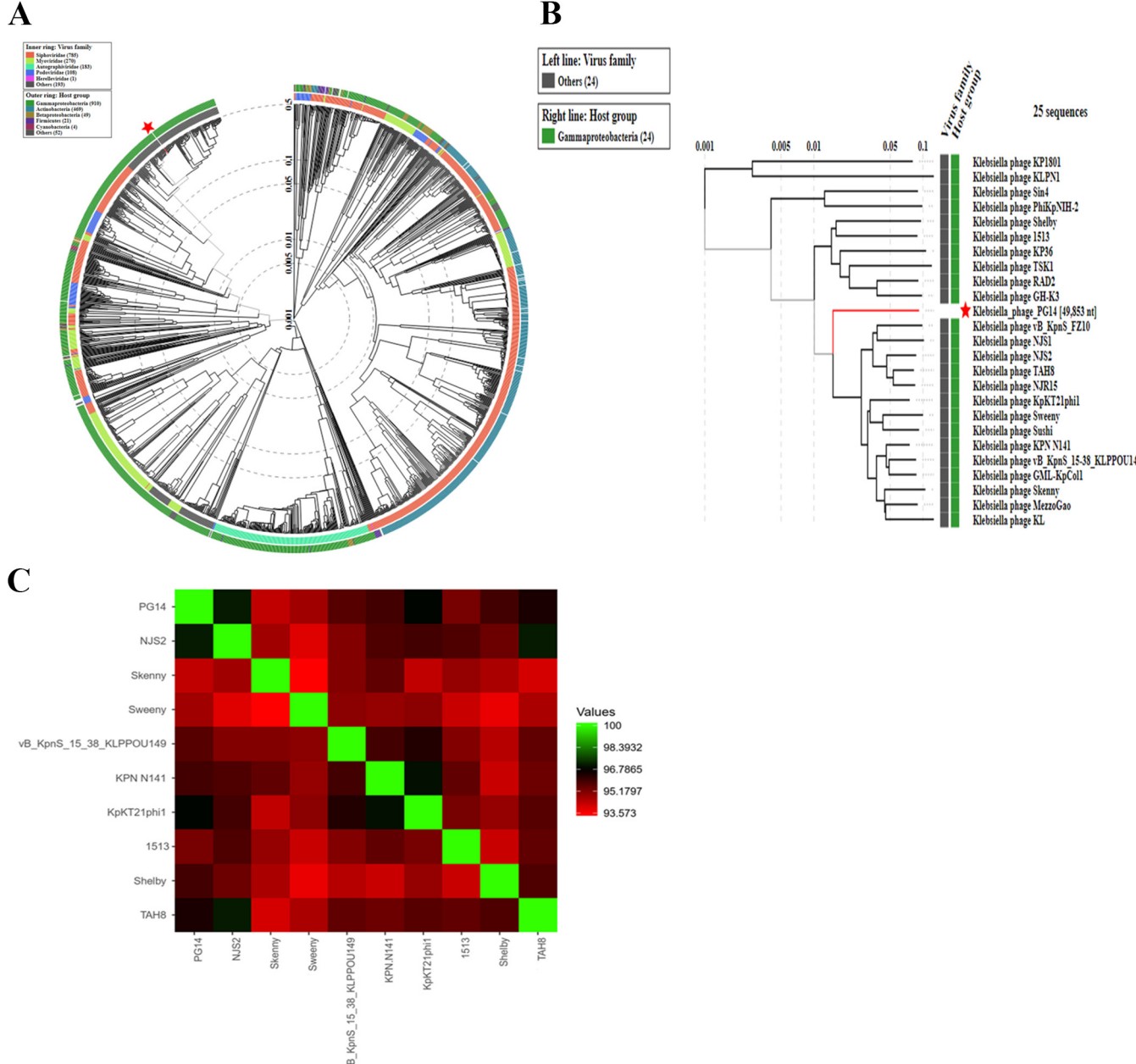

**FIG 3** (A) Proteomic tree comparing the whole genome amino acid sequences of phage PG14 and showing its relatedness with other phage whole genome amino acid sequences. (B) Enlarged view of part of proteomic tree comparing the whole genome amino acid sequences of phage PG14 and showing its relatedness with 24 other phage whole genome amino acid sequences. (C) Heat map of Ortho ANI values of *Klebsiella* phage PG14 and closely related phages. The values were calculated using OAT software, and a heat map was generated by the heat mapper online tool.

compared to proteinase K and DNase I treatment. The result indicated that polysaccharide was the major component of the *K. pneumoniae* G14 biofilm matrix. The expression of multiple depolymerases in phage tail fibers generally disrupts polysaccharides. The phage PG14 genome has shown the presence of putative genes encoding depolymerases, which can be related to the antibiofilm activity of phage PG14.

**(ii) Biofilm inhibition.** The phage PG14-inhibited biofilms formed by *K. pneumoniae* G14 showed an 80% reduction in attached biomass. No significant difference was observed in the attached biomass of *K. pneumoniae* G14 treated with phage PG14 at different MOI when assessed by the crystal violet assay (CV assay). However, a 0.5 to 5.5 log reduction in viable cell count was observed in samples treated with phage G14 at each MOI compared to the control ($P < 0.05$) (Fig. 5).

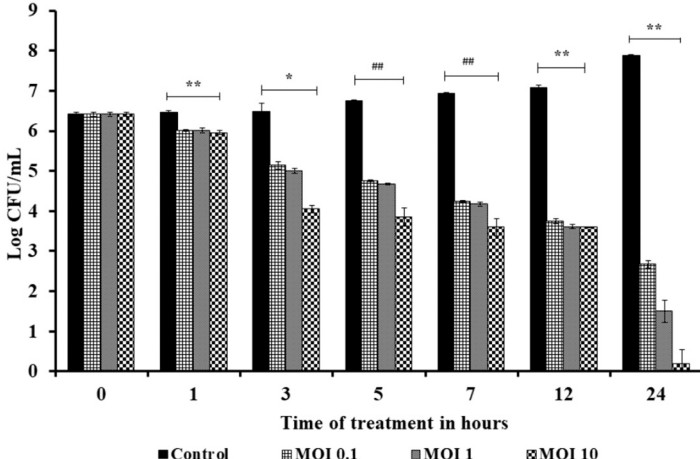

**FIG 4** Treatment of planktonic cells of *K. pneumoniae* G14 with *Klebsiella* phage PG14 at 0.1, 1, and 10 MOI. After treatment for 1, 3, 5, 7, 12, and 24 h, the viable cell counts were determined. Data represent the means $\pm$ SD of three independent experiments. Bars with an asterisk and hash are statistically different from the untreated control. According to the unpaired $t$ test for different variance, *, $P < 0.01$; ##, $P < 0.005$; and **, $P < 0.001$.

The confocal laser scanning microscopy (CLSM) analysis of phage PG14-treated *K. pneumoniae* G14 showed total biofilm inhibition (BI) resulting in dead biomass (red fluorescence). Both dead and live biomass present in the biofilm matrix was estimated in a CV assay based on biomass staining using the dye. However, dead biomass was eliminated when the live cell count was taken, which could explain the difference observed in the CV and viable cell count assay results (Fig. 6).

**(iii) Biofilm disruption.** The phage PG14 disrupted the preformed biofilms of *K. pneumoniae* G14 and other phage-susceptible host bacteria showing up to 71% disruption in attached biomass. The CV assay showed no significant difference in the attached biomass of *K. pneumoniae* G14 treated with different concentrations of phage PG14. However, a 1.3 to 6.3 log reduction in viable cell count was observed in preformed biofilm treated with different concentrations of phages compared to the control ($P < 0.05$) (Fig. 7).

The field emission scanning electron microscopic (FESEM) images also confirmed the total disruption of preformed biofilms after 24 h treatment with phage PG14 as observed by the reduced cell number and distorted morphology of the cells within the biofilm

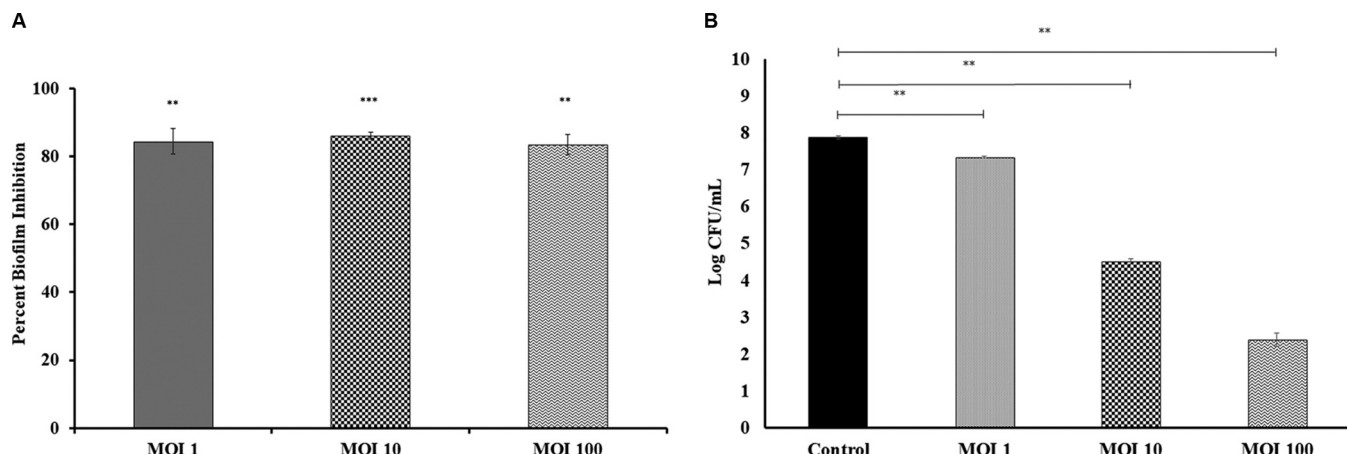

**FIG 5** To check the biofilm inhibition effect of phage PG14, *K. pneumoniae* G14 was coincubated with phage PG14 at 1, 10, and 100 MOI for 24 h at 37°C. After incubation, biofilm formed of *K. pneumoniae* G14 was quantified for treated and control samples. (A) Results of adhered cell biomass quantified by CV assay (absorbance at 595 nm). (B) Results of viable cell counts of biomass determined as CFU per milliliter. Data represent the means $\pm$ SD of three independent experiments. Bars with an asterisk are statistically different from the untreated control. According to the unpaired $t$ test for different variance, $P < 0.001$ (**) and $P < 0.0001$ (***) were considered statistically significant.

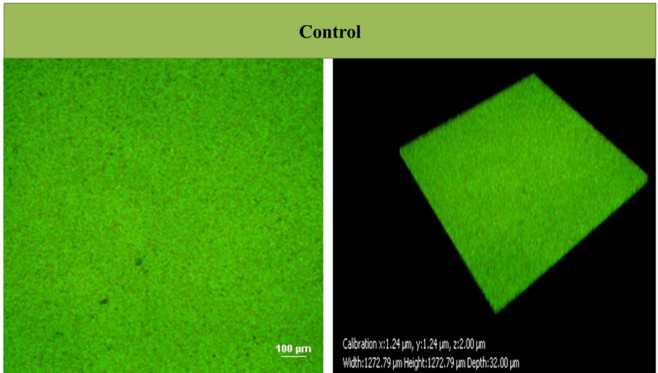
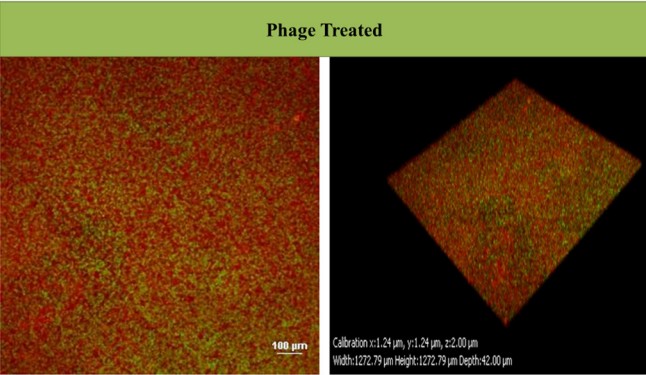

**FIG 6** To check the biofilm inhibition effect of phage PG14, *K. pneumoniae* G14 was coincubated with phage PG14 (1 × 10$^8$ PFU/mL) for 24 h at 37°C. After incubation, *K. pneumoniae* G14 biofilm was visualized and quantified by confocal laser scanning microscopy (CLSM) for both treated and control samples. For visualization, the adhered cell biomass was stained by LIVE/DEAD BacLight bacterial viability kit as per the manufacturer's protocol (Invitrogen, CA, USA). Samples were observed under a confocal laser scanning microscope (Nikon A1 R) using a 20× objective. Filters with an excitation wavelength of 488 nm and 588 nm were used to observe the fluorescence for both live (green) and dead (red) bacteria.

(Fig. 8). In addition, the confocal laser scanning microscopy (CLSM) analysis revealed that the number of dead or compromised cells (red fluorescence) increased with increasing treatment time, demonstrating a gradual disruption of *K. pneumoniae* G14 biofilms treated with phage PG14 (Fig. 9). The measurement of fluorescent intensities revealed that 99% of biofilm was disrupted within 7 h. However, disruption after 24 h treatment was 74%, suggesting possible regrowth of surviving cells within the biofilm.

## DISCUSSION

The emergence of *K. pneumoniae* resistant to multiple antibiotics presents a grave threat to global public health. For effective treatment of *K. pneumoniae* infections, it becomes advantageous to inhibit and disrupt the biofilm produced by pathogenic *K. pneumoniae*. Considering the increasing ineffectiveness of antibiotics, the use of phage therapy to eliminate recalcitrant *K. pneumoniae* infections seems promising (17, 26, 27). In this study, we have isolated and characterized a phage named *Klebsiella* phage PG14.

It is necessary to characterize an individual phage thoroughly when it is considered for customized phage therapy. After characterization, the phage PG14 showed a turbid halo zone around the clear circular plaques, indicating polysaccharide depolymerase activity or activity of lysin proteins diffused in the surrounding area (28). The TEM characterization of phage PG14 confirmed its resemblance with phages belonging to *Siphoviridae*, the most abundant family of bacteriophages (29). The stability of phage PG14 at alkaline pH suggests its potential use as a therapeutic agent against UTI. The success of phage therapy is generally attributed to factors like adsorption rate and burst size (30). The relatively higher adsorption rate and the large burst size in the case of phage PG14 indicate its higher optimum lysis, thereby suggesting its potential as an effective therapeutic agent.

Usually, to ensure the fitness of phages as antibacterial agents, it becomes necessary to carry out a thorough genotypic analysis in addition to the phenotypic characterization. The phages with antibiotic resistance or bacterial virulence genes in their genome may transfer these genes between their bacterial hosts used for large-scale production or to which phage can infect. Hence, the presence of antibiotic resistance or bacterial virulence genes is considered an unsuitable property of phage preparation (15, 31). The whole-genome sequence of phage PG14 does not carry any putative genes associated with virulence or/ and antibiotic resistance. The ability of bacteriophages to infect multiple hosts is correlated with the expression of multiple depolymerases present in phage tail fibers and is generally involved in the disruption of polysaccharides and, thereby, in the adsorption on the host cell (32). In our study, the prediction of multiple open reading frames (ORFs) associated with tail fiber proteins and halo zones observed around the clear plaques explains the ability of phage PG14 to lyse *K. pneumoniae*, *P. aeruginosa*, and *E. coli* isolates. Additionally, the

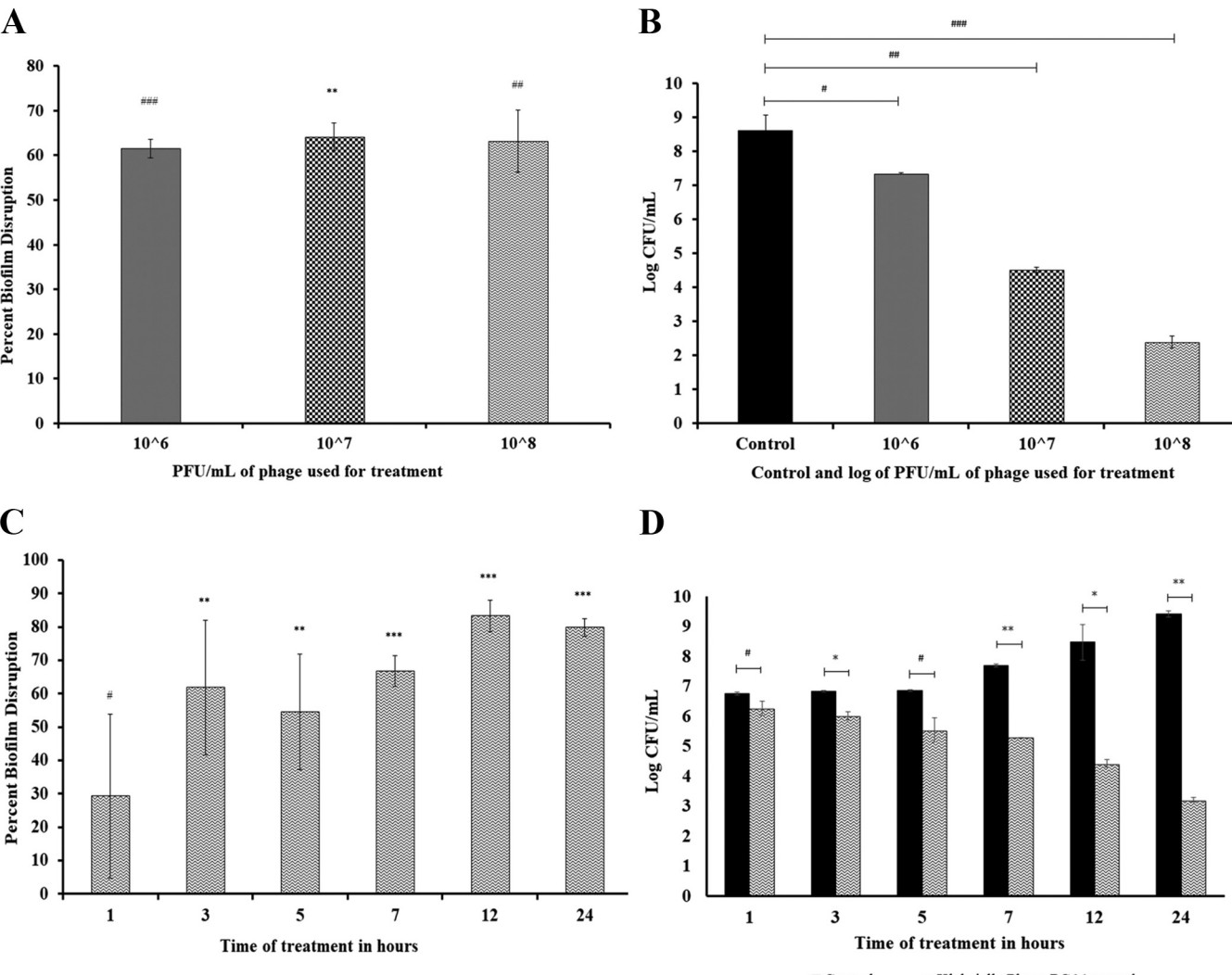

**FIG 7** Effect of phage PG14 on preformed biofilm, estimated by CV assay and by a total viable count of adhered biomass. Biofilm of *K. pneumoniae* G14 was allowed to form for 24 h. After 24 h, planktonic cells were removed, and the remaining adhered cells were treated with phage PG14. In one set of the experiment, the treatment of phage PG14 was given at different concentrations of phage ($1 \times 10^6$, $1 \times 10^7$, and $1 \times 10^8$ PFU/mL) and then incubated at 37°C. In another set of experiments, treatment was given for different time intervals (1, 3, 5, 7, or 24 h) of incubation, and untreated controls were kept in all of the experiments. After treatment, the adhered biofilm was quantified by CV assay (absorbance at 595 nm). (A) Result of the CV assay for different concentration of phage. (C) results for treatment given for different time interval quantified by the CV assay. Biofilm biomass was also quantified by viable cell counts, determined as CFU/mL. (B) Results of viable cell count for different concentrations of phage. (D) Results for treatment given for different time intervals quantified by viable cell count. Data represent the means ± standard deviations of three independent experiments. Bars with an asterisk are statistically different from the untreated control. According to the unpaired *t* test for different variance, $P < 0.05$ (#), $P < 0.01$ (*), $P < 0.005$ (##), $P < 0.001$ (**), $P < 0.0005$ (###), and $P < 0.0001$ (***) were considered statistically significant.

putative host lysis protein genes predicted during the *in silico* analysis can be used to develop future therapeutics like enzybiotics.

The biofilm produced by *K. pneumoniae* provides physical protection against the host immune response and prevents antibiotic penetration (27). The major components in bacterial biofilm matrix generally include polysaccharides, protein, and DNA (33, 34). The phage tail fibers contain depolymerase activity, which can disrupt the polysaccharides. Phages have also been reported to act as antibiofilm agents against many MDR pathogens, including *K. pneumoniae* (32, 35, 36). Phage PG14 showed a significant reduction in the *K. pneumoniae* G14 planktonic cells. Additionally, the phage PG14 showed the ability to inhibit biofilms and disrupt preformed biofilms of *K. pneumoniae*, which corroborates with previously reported phages (17, 28). The experiments carried out in this study, including total viable count (TVC) assay and confocal microscopic analysis have confirmed the ability of phage PG14 to inhibit and disrupt the biofilm produced by *K. pneumoniae* significantly.

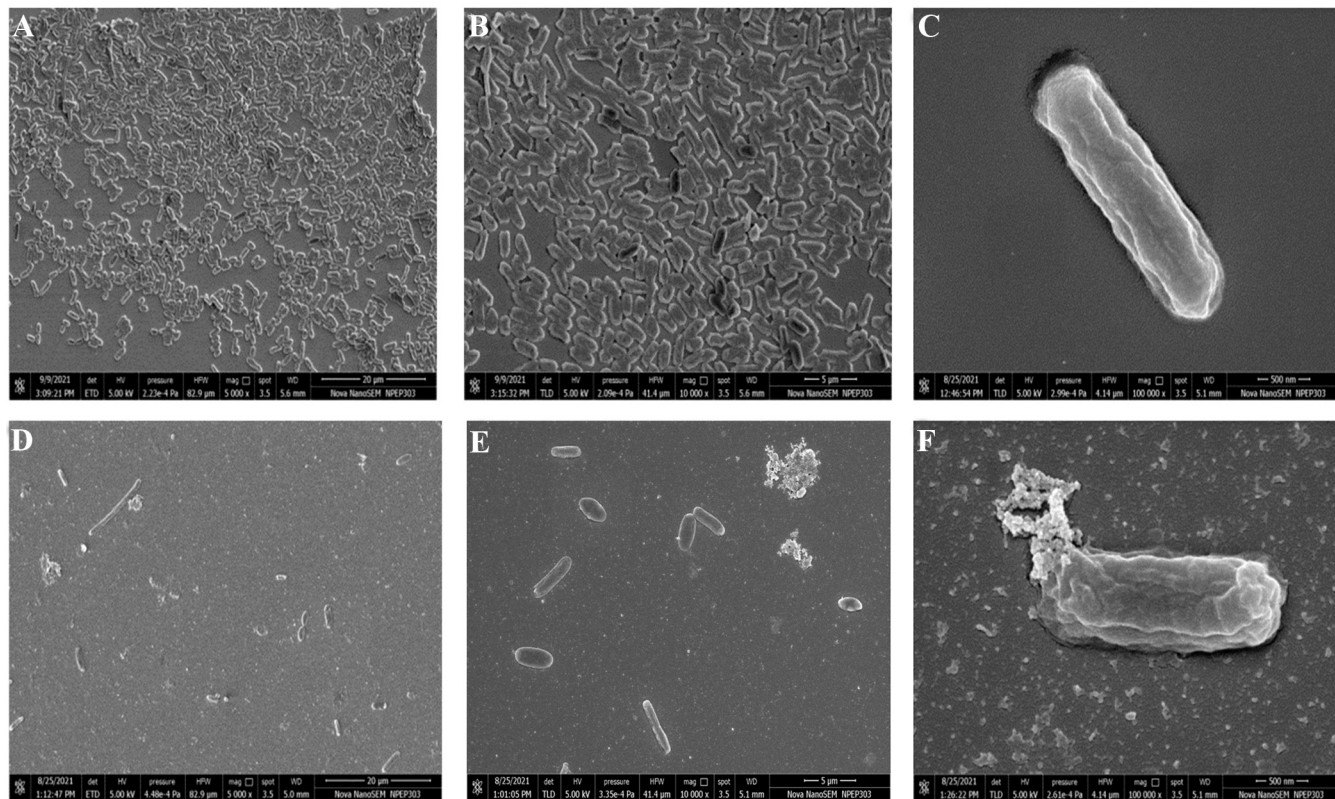

**FIG 8** Effect of phage PG14 on preformed biofilm of *K. pneumoniae* G14 visualized by field emission scanning electron microscopy (FESEM). The biofilm of *K. pneumoniae* G14 was allowed to form on 5-mm by 5-mm glass slides for 24 h. Biofilm formed after incubation was treated with phage PG14 (1 × 10⁸ PFU/mL) for 24 h. After treatment, the adhered cell biomass was fixed with glutaraldehyde, coated with gold, and observed under FESEM at 5,000×, 10,000×, and 100,000× magnifications. (A, B, and C) Images of control biofilm at different magnifications; (D, E, and F) images of phage PG14-treated biofilm at different magnifications.

Similar to the study by Townsend et al. (18), the phage PG14 was able to suppress *K. pneumoniae* G14 biofilm growth for up to 12 h and showed minor regrowth at 24 h, although the number of viable cells in phage-treated biofilm was significantly less than in the control. The practical application of phage therapy in treating infections associated with biofilms needs a comprehensive understanding of the phage-host interactions and the regulation of phage resistance in their corresponding bacterial hosts (36). The emergence of bacteriophage-resistant clones can be the reason for this phenomenon, as bacteria can become resistant to phages over infection; hence, further investigation is required to validate this phenomenon. Alternative strategies to overcome the problem of phage resistance clones could be using phages in preparations, such as a phage cocktail, a combination of phages and antibiotics, or genetically modified phages that can be tried for treating bacterial infections (18, 23, 37, 38). Phage PG14 illustrated a lytic ability covering a variety of bacterial pathogens, including *K. pneumoniae*, *P. aeruginosa*, and *E. coli*, which all can cause nosocomial infections. One of the limitations of this study is that the lytic ability of the phage PG14 was tested against only 13 bacterial isolates, which can be further explored for more bacterial isolates. Phages will not likely replace antibiotics entirely owing to a few limitations, such as narrow specificity, development of phage resistance by bacteria, and bacterial lysis-related effects. However, these limitations can be overcome by using phage cocktails, genetic engineering of phages, or phage products like enzybiotics (39, 40). Our results suggest that phage PG14 can be a beneficial addition to the *Klebsiella* phages with therapeutic potential to control *K. pneumoniae* biofilms.

## MATERIALS AND METHODS

**Bacterial strains.** The clinical isolates of *K. pneumoniae* (*n* = 4), *Staphylococcus aureus* (*n* = 4), *Acinetobacter baumannii* (*n* = 2), *Escherichia coli* (*n* = 1), and *Pseudomonas aeruginosa* (*n* = 2) used in this

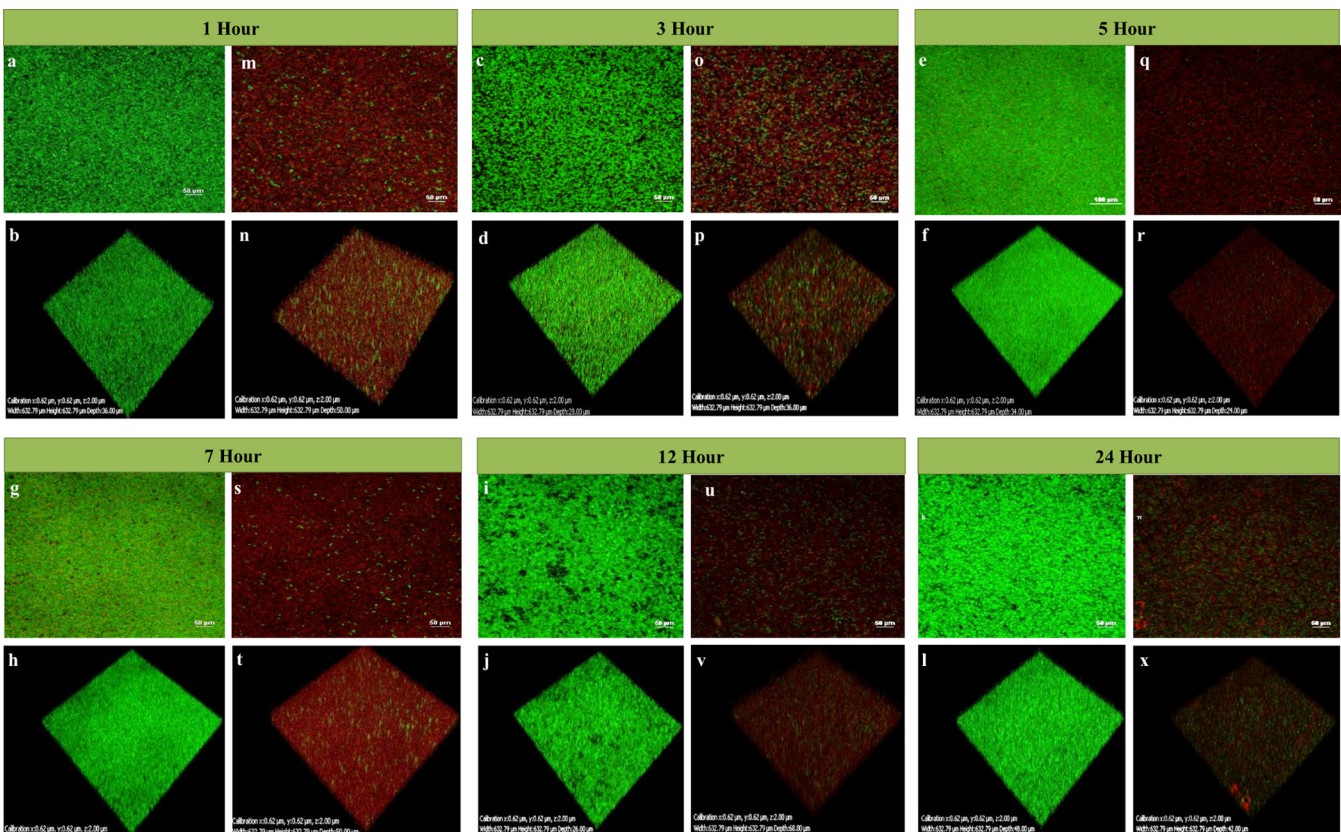

**FIG 9** Effect of phage PG14 on preformed biofilm of *K. pneumoniae* G14 visualized and quantified by confocal laser scanning microscopy. The biofilm of *K. pneumoniae* G14 was allowed to form for 24 h. After 24 h of incubation, planktonic cells were removed, and the remaining adhered cells were then treated with phage PG14 (1 × 10⁸ PFU/mL) at different time intervals, and untreated controls were kept for all the time intervals taken. The adhered cell biomass in both treated and control samples was stained by LIVE/DEAD BacLight bacterial viability kit as per the manufacturer's protocol (Invitrogen, CA, USA). Samples were observed under a confocal laser scanning microscope (Nikon A1 R) using a 20× objective. Filters with an excitation wavelength of 488 nm and 588 nm were used to observe the fluorescence for both live (green) and dead (red) bacteria. The results for control and treatment for different time intervals are represented in the panels as follows: control (A, B) and treated for 1 h (M, N), control (C, D) and treated for 3 h (O, P), control (E, F) and treated for 5 h (Q, R), control (G, H) and treated for 7 h (S, T), control (I, J) and treated for 12 h (U, V), and control (K, L) and treated for 24 h (W, X).

study are a part of the laboratory collection of clinical isolates obtained from various hospitals across India. Confirmation of the phenotypic identity of isolates was performed using the VITEK 2 system (bioMérieux, Inc., USA), and the resistance profiles were studied as recommended by the Clinical and Laboratory Standards Institute (CLSI). All of the isolates were preserved in 40% glycerol phosphate buffer saline (PBS) (137 mM NaCl, 2.7 mM KCl, 10 mM $Na_2HPO_4$, 2 mM $KH_2PO_4$ [pH 7.4]) stocks at −80°C. When required, the bacterial isolates were cultivated in lysogeny broth (LB) (HiMedia, India) at 37°C with shaking conditions at 120 rotation per minute (RPM).

**Isolation, enrichment, and purification of *Klebsiella* phage PG14.** *K. pneumoniae* G14 was used as a primary host for isolating phages against *Klebsiella* spp. Water samples collected from the Mutha River, Pune, Maharashtra, India, were used as a source to isolate phages. The initial enrichment, screening, and purification of phages were done as described previously, and based on the results, the bacteriophage *Klebsiella* phage PG14 was selected for further studies (41).

Briefly, water samples were filtered through a coarse filter and centrifuged at 8,000 rpm for 20 min at 4°C to remove any particulate matter. The supernatant obtained was filtered through a 0.22-$\mu$m membrane syringe filter (Axiva Sichem Biotech, India) to remove bacterial cells, and the filtered lysate was stored at 4°C.

Fifty milliliters double-strength phage broth (tryptone, 10.0 g/L; yeast extract, 5.0 g/L; peptone 5.0 g/L; dipotassium hydrogen phosphate, 5.0 g/L; pH, 6.8 ± 0.2) was inoculated with 1 mL of exponential phase culture of *K. pneumoniae* G14 (~1 × 10⁶ CFU/mL) and incubated at 37°C for 24 h, followed by which 50 mL of previously prepared lysate was added and further incubated at 37°C for 24 h to obtain the enriched phage lysate. Confirmation of phages in the enriched lysate was done by spot inoculating the phage lysate to observe for a clearance of bacterial lawn growing on the lysogeny agar plate. The enumeration of phages in the enriched lysate was done using the soft agar overlay method. For purification, a single well-isolated plaque was picked using a micropipette tip, inoculated in 5 mL of phage broth along with exponential phase culture of *K. pneumoniae* G14 and enriched for 24 h. The multiplicity of infection (MOI) was calculated using the following formula: plaque forming units/mL divided by CFU/mL (PFU/CFU).

High-titer preparation of purified phages was mixed with 10% polyethylene glycol 6000 (PEG 6000) and 1 M NaCl followed by incubation at 4°C for 24 h to allow precipitation of phages. After precipitation, the phage particles were centrifuged at 14,000 × *g* for 60 min to obtain a pellet that was resuspended in sterile SM buffer (for 1 L, 5.8 g NaCl, 50 mL 1 M Tris-HCl [pH 7.5], 2 g MgSO₄·7H₂O, and 5 mL 2% gelatin). The SM buffer containing phage particles was dialyzed against PBS up to 6 h with intermittent changing of buffer every 2 h to remove the salts and other impurities. The purified phage suspension was stored at 4°C for further analysis (19, 21, 42).

**Characterization of the bacteriophage *Klebsiella* phage PG14. (i) Phage adsorption rate and one-step growth curve.** Adsorption rate assay was performed as described previously with minor modifications (19, 21). Exponentially grown *K. pneumoniae* G14 culture was mixed with the phage PG14 at an MOI of 0.001 and incubated at 37°C. Aliquots from the above mixture were removed at 4-min intervals up to 40 min and filtered using 0.22-$\mu$m membrane filters (Axiva Sichem Biotech, India) to remove adsorbed phages. Subsequently, the filtrate was titrated with the host bacterium to determine the number of nonadsorbed (free) phages using the soft agar overlay method. The adsorption curve was constructed by plotting the percentage of free phages in the solution versus time.

For the one-step growth curve, 10 mL of exponential phase culture of *K. pneumoniae* G14 ($\sim$1 × 10⁷ CFU/mL) was centrifuged at 8,000 rpm for 15 min at 4°C. Tryptic soy broth (TSB) (HiMedia, India) and phage lysate were added to the pellet to achieve an MOI of 0.01. The mixture was incubated at 37°C for 10 min and centrifuged at 8,000 rpm for 15 min at 4°C to remove the nonadsorbed (free) phages. The pellet containing the adsorbed phages was resuspended in 10 mL TSB. Aliquots were taken at 0, 10, 20, 30, 40, 50, 60, and up to 120 min, and phage titer was determined using the soft agar overlay method. A graph of time versus log PFU/mL was plotted to determine the phage's latent stage and burst size.

**(ii) Determination of host range and efficiency of plating.** Spot test method as described above was used to determine the host range of phage PG14 against 13 bacterial isolates. The six bacterial isolates showing sensitivity to phage PG14 were selected to determine EOP using the soft agar overlay method. The EOP was calculated as the ratio between PFU per milliliter on a sensitive strain and PFU per milliliter on the primary host (*K. pneumoniae* G14) and then ranked as "high productive" ($\geq$0.5), "medium productive" (0.1 $\leq$ EOP < 0.5), "low productive" (0.001 < EOP < 0.1), and "inefficient" (<0.001) (43, 44).

**(iii) Influence of physical agents on phage viability.** The stability of phage PG14 was checked at different pH and temperatures by the method described earlier. Briefly, phages were suspended at approximately 10⁸ PFU/mL in 1 mL SM buffer, previously adjusted with 1 M NaOH or 1 M HCL, to yield a pH range from 1 to 14. The sample tubes were incubated at 30°C for 60 min. Serial dilutions of samples from each tube with different pH were titrated against *K. pneumoniae* G14 by the soft agar overlay method. The phage stability at different temperatures was determined by incubation of phage PG14 at 10, 20, 30, 40, 50, 60, 70, and 80°C for 60 min. Serial dilutions of each sample were titrated against *K. pneumoniae* G14 by the soft agar overlay method. Experiments were carried out in triplicates, and log PFU/mL counts were plotted against different pH and temperature values (41).

**(iv) Transmission electron microscopy.** The phage PG14 ($\sim$1 × 10⁸ PFU/mL) sample was fixed using 2.5% glutaraldehyde and 4% paraformaldehyde (PFA). Five microliters of the PFA fixed sample was placed on a coated copper grid, allowed to adsorb on the surface for 5 min, and excess of the sample was removed. The grid was floated with the sample side down on a droplet of 2% uranyl acetate stain for 5 min. The excess stain was removed, and the grid was dried under a table lamp. Sections were observed under JEOL 1400 plus transmission electron microscopy (TEM) (Japan) at 120 kV, and microphotographs were captured using a charge-coupled-device (CCD) camera (Olympus, Münster, Germany). Morphological classification and naming of phage were done as per the recommended guidelines (29, 45–47).

**Phage DNA extraction.** DNA was isolated from phage PG14 using the phenol-chloroform method with slight modification. Briefly, the lysate was treated with 10 $\mu$L of each DNase I (1 U/$\mu$L) and RNase A (4 mg/mL) and incubated for 45 min at 37°C to remove any bacterial nucleic acid remaining in the lysate, followed by which the reaction was stopped by placing the tube at 75°C in a water bath for 15 min. To release DNA out from the protein coat, 10 mM EDTA, 50 $\mu$L of 10% SDS, and 5 $\mu$L of 20 mg/mL of proteinase K were added to the above mixture and incubated at 56°C for 2 h. After incubation, the mixture was allowed to cool and deproteinized using phenol-chloroform (24:1) extraction. The DNA in the aqueous phase was precipitated in the next step using 70% ethanol. Finally, the DNA pellet obtained after centrifugation was air-dried and resuspended in 50 $\mu$L Tris-EDTA (TE) buffer (19, 48, 49).

**Phage genome sequencing and analysis.** The quantity and purity of the DNA sample were checked using the DeNovix DS-11 spectrophotometer. The purified DNA was used to prepare the libraries using the Nextera XT kit (2 × 150 bp chemistry) following the manufacturer's protocol. The obtained reads were checked by FastQC, and reads with a phred score of $\geq$30 and without adapters were selected to generate *de novo* assembly using SPAdes v.3.13.1.

The assembly generated was validated based on sequence homology to known phage sequences in NCBI via BLASTN. The putative proteins in the genome were predicted by the RAST annotation scheme (https://rast.nmpdr.org). Phage PG14 genome map was constructed using the tool GenomeVx (http://wolfe.ucd.ie/GenomeVx/). The proteomic tree, based on the whole-genome amino acid sequences of *Klebsiella* phage PG14 and closely related phages, was generated using VipTree (https://www.genome.jp/viptree) (50). A heat map of OrthoANI values of phage PG14 and other closely related phages was calculated using the OrthoANI tool version 0.93.1 (https://www.ezbiocloud.net/tools/orthoani), and a heat map was generated by using a heat mapper (http://www.heatmapper.ca).

***In vitro* efficacy testing of phage PG14. (i) Time-kill assay for planktonic cells.** Time kill assay of *K. pneumoniae* G14 was performed to check *in vitro* killing efficiency of phage PG14. For this study, host culture was grown in 50 mL of LB broth at 37°C ($\sim$1 × 10⁶ CFU/mL) and infected with phage PG14 with

an MOI of 0.01, 0.1, and 1. The content was incubated at 37°C with shaking condition at 120 RPM. One-milliliter aliquots of the sample were removed at 0, 1, 3, 5, 7, 12, and 24 h. Each aliquot was centrifuged at 13,000 $\times$ *g* for 2 min. The obtained bacterial pellet was resuspended in saline, serially diluted, and plated on LB agar plates to obtain CFU per milliliter. Untreated control samples were also run to obtain the bacterial count. Experiments were carried out in triplicates (41, 51).

**(ii) Biochemical characterization of the biofilm matrix.** Biochemical characterization of the biofilm matrix was done as described earlier (52). Briefly, 24-h-old *K. pneumoniae* G14 biofilms were washed with PBS and then treated separately for 1 h at 37°C either with a solution of 10 mM sodium metaperiodate (HiMedia, India) (used to disrupt the extracellular polysaccharides), with 100 $\mu$g/mL of proteinase K (Genei Laboratories, India) (used to disrupt the extracellular proteins), or with 100 units/mL of DNase I (New England Biolabs) (used to disrupt the extracellular DNA). After treatments, the biofilms were quantified by CV assay (37, 52).

**(iii) Biofilm inhibition assay.** Biofilm inhibition by phage PG14 was analyzed by CV assay to determine biofilm mass in the control biofilm and PG14-treated biofilm. Biofilm formation and treatment were performed as previously described with slight modifications (37, 53). Briefly, overnight grown cultures of host bacterium were diluted in fresh TSB medium to achieve an approximate density of $1 \times 10^6$ CFU/mL and inoculated into a 24-well polystyrene microtiter plate (Tarson, India) and incubated for 24 h at 37°C. For biofilm inhibition assay, the wells containing host bacterium were treated immediately with phage PG14 with different MOI (1, 10, and 100), and plates were incubated at 37°C for 24 h; untreated wells were kept as control. To measure the effectiveness of the different treatments, the total biomass of biofilm in control and treated wells was quantified by performing the CV assay described previously (37). Briefly, 24-h-old biofilms were washed with PBS and stained with 1 mL of 0.1% (wt/vol) crystal violet for 15 min. After staining the biofilm, the excess CV was washed twice with water and air-dried. Finally, the dye was solubilized by adding 33% (vol/vol) acetic acid, and the absorbance at 595 nm was measured using a microplate reader (SpectraMax M2e; Molecular Devices, USA).

Biofilm Inhibition of bacterial hosts by phage PG14 was also analyzed to determine the number of viable cells in the biofilm and was quantified by the method described previously (37). Briefly, biofilms were scraped and suspended in PBS. Ten-microliter droplets from 10-fold serial dilutions of this cell suspension were spotted onto LB agar plates and incubated at 37°C for 24 h. The cell counts obtained in these experiments were used to determine the number of CFU per milliliter.

**(iv) Biofilm disruption assay.** Biofilm disruption was analyzed using a CV assay. As described above, biofilm was formed in a 24-well polystyrene microtiter plate (Tarson, India). For biofilm disruption assay, biofilm was allowed to be formed for 24 h; the planktonic cells were removed by washing with PBS. The biofilms were washed twice with PBS and then treated with phage PG14 ($1 \times 10^6$, $1 \times 10^7$, and $1 \times 10^8$ PFU/mL) and incubated at 37°C. In other set of experiment, treatment was given for 1, 3, 5, 7, 12, and 24 h with phage PG14 ($1 \times 10^8$ PFU/mL). Total biomass was quantified by the CV assay described above.

Biofilm disruption by phage PG14 was also analyzed by determining the TVC count of bacterial cells in the control biofilm and the PG14-treated biofilm. The preformed biofilm was treated with phage PG14 ($1 \times 10^6$, $1 \times 10^7$, and $1 \times 10^8$ PFU/mL) and incubated at 37°C. In other set of experiment, treatment was given for 1, 3, 5, 7, 12, and 24 h with phage PG14 ($1 \times 10^8$ PFU/mL), and untreated biofilm was kept as control. The number of viable cells in the biofilm biomass was quantified by the method described above.

**(v) Confocal laser scanning microscopy.** For confocal microscopic analysis, 24-h-old biofilms were formed in a 24-well polystyrene microtiter plate (Tarson, India) and treated for 1, 3, 5, 7, 12, and 24 h by the method described above. After treatment, wells were washed twice with PBS and stained using the LIVE/DEAD BacLight bacterial viability kit as per the manufacturer's protocol (Invitrogen, CA, USA). Samples were observed under a confocal scanning laser microscope (Nikon A1 R) using a 20× objective, and the fluorescence from both live (green) and dead (red) bacteria were observed using filters with an excitation wavelength of 488 nm and 588 nm (37, 54).

**(vi) Field emission scanning electron microscopic imaging of biofilm.** Twenty-four-hour-old biofilms were formed by inoculating 1 mL of host bacterial cell suspension containing approximately $10^6$ CFU/mL in TSB on glass slides (5-mm by 5-mm cut pieces), and after growth, the planktonic cells were removed and the biofilm was washed twice with PBS. Biofilm treated with phage PG14 ($1 \times 10^8$ PFU/mL) and untreated control was incubated at 37°C for 24 h. After treatment, samples were prepared by the earlier protocol (55). Briefly, biofilms were gently washed with PBS twice, fixed with 2.5% glutaraldehyde, and kept overnight at 4°C, followed by a PBS wash. Dehydration was done sequentially with grades of 20, 40, 60, 80, and 90% ethanol for 15 min each and then twice with absolute ethanol. Samples were dried, coated with gold, and observed under FESEM (Nova NanoSEM NPEP 303).

**Statistical analysis.** For statistical analysis, the unpaired *t* test for different variances was used. Differences were considered statistically significant at a *P* value of <0.05.

**Data availability.** The genome sequence data of phage PG14 has been submitted to the National Center for Biotechnology Information (NCBI) (https://www.ncbi.nlm.nih.gov/nuccore/) under accession number OM964875.

## ACKNOWLEDGMENTS

We thank the Central Instrumentation Facility at SPPU, Pune, India, for the FESEM and CLSM facility and Tata Memorial Centre Advanced Centre for Treatment, Research and Education in Cancer, Navi Mumbai, India, for the TEM facility.

We thank Aishwarya Patil and Shripad Nimbalkar, postgraduate dissertation students, for their initial help in the study and Suresh Basutkar for collections of cultures.

We declare no known conflict of interest.

Conceptualization, K.R.P. and M.S.M.; Methodology, K.R.P. and M.S.M.; Validation, K.R.P., M.S.M., and S.N.K.; Formal Analysis, K.R.P., M.S.M., and S.N.K.; Investigation, M.S.M.; Resources, K.R.P.; Data Curation, M.S.M.; Writing – Original Draft Preparation, M.S.M.; Writing – Review and Editing, K.R.P., M.S.M., and S.N.K.; Visualization, M.S.M.; Supervision, K.R.P.; Project Administration, K.R.P.; Funding Acquisition, K.R.P.

All authors have read and agreed to the published version of the manuscript.

This research was funded by Rashtriya Uchchatar Shiksha Abhiyan (RUSA-TH3.2), the Department of Higher Education, Ministry of Education, Government of India. S.N.K. is a recipient of the CMSRF-2019 fellowship provided by SARATHI, Pune.

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
