## [Reviewer comments · Microbiology Spectrum]

Microbiology Spectrum

Characterization of novel Klebsiella phage PG14 and its antibiofilm efficacy

Mansura Mulani, Shital Kumkar, and Karishma Pardesi

Corresponding Author(s): Karishma Pardesi, Savitribai Phule Pune University, Pune, Maharashtra, India

Review Timeline:

Submission Date:	May 28, 2022
Editorial Decision:	August 17, 2022
Revision Received:	October 3, 2022
Accepted:	October 19, 2022

Editor: Daria Van Tyne

Reviewer(s): Disclosure of reviewer identity is with reference to reviewer comments included in decision letter(s). The following individuals involved in review of your submission have agreed to reveal their identity: Jessica C. Sacher (Reviewer #1); Eleanor Jameson (Reviewer #2)

Transaction Report:

DOI: <https://doi.org/10.1128/spectrum.01994-22>

August 17, 2022

Dr. Karishma Rajendra Pardesi
Savitribai Phule Pune University, Pune, Maharashtra, India
Department of Microbiology
Ganeshkhind road
Pune, Maharashtra 411007
India

Re: Spectrum01994-22 (Characterization of novel Klebsiella phage PG14 and its antibiofilm efficacy)

Dear Dr. Karishma Rajendra Pardesi:

Thank you for submitting your manuscript to Microbiology Spectrum. Your manuscript was reviewed by two experts and I would now like you to revise your study in line with their feedback. Please address all comments in your revised submission. In addition, please include a "data availability" statement in the Methods section with the accession numbers of nucleotide sequence data generated in this study.

Link Not Available

Sincerely,

Daria Van Tyne

Journals Department
Reviewer comments:

Reviewer #1 (Comments for the Author):

Summary: The manuscript contains useful data and represents a thorough characterization of the phage. It also has a nicely detailed methods section. However, the presentation of results sections (figures, figure legends, results) needs work.

Main issues:

Results section headings are generally not descriptive of the text, especially first two - I suggest rewriting them to emphasize the specific results. In addition, some results sections (body text) lack detail (one is only one sentence long).

Most figure legends need more detail about what was done. Preferably the authors should get help with figure legend and figure conventions, and with stats.

Figures are almost all blurry/hard to read, need work getting to publication quality.

Examples of issues with figures and figure legends:

- Aa / Ab / Bb naming in figs is confusing

Table 1

- EOP usually a quantitative value: what does each qualitative result correspond to? Should appear in legend. Would actually suggest a quantitative assessment if reporting as EOP

Fig 1 A - difficult to see what's going on in this image. B - can't see plaque morphology due to angled blurry image

TEM - which phage is the one? many smaller heads too - looks like mixed sample

- temperature - should indicate how long treatment was

Fig 4

- why mixed use of ## and * instead of ** / ***

Fig 5: legend specifically needs work (grammar, syntax, punctuation, capitalization)

fig 6 - should give more detail in legend about microscopy parameters used

Fig 7 - check X axis labels, e.g. 'Control and log of PFU/ml of phage used for treatment..'

Fig 7 axis labels are inconsistent sizes, some blurry etc. Inconsistent labeling, ie. Fig7A is 'Time in Hour' and Fig 7B is 'Time of Treatment in Hours'

Fig 8 legend: run-on sentence, unclear

Results and discussion:

Host range: it would be nice to know why the strains chosen for host range were chosen (both in the text and in table 1).

Section: 'Antibiofilm activity of phage PG14'

- 'polysaccharide was its major component'. What kind of polysaccharide? how was this determined?

- very sparse section, need details on biochemical characterization mentioned

Discussion

- Very long run-on paragraphs

- check wording: 'to the suite of the basis for the assessment'

Writing:

The writing has several issues with sentence structure, punctuation, word choice, capitalization, inconsistencies, etc

Examples:

- switch from S to +

- check comma usage throughout

- check definition of opaque

- define acronyms TEM, EOP, SEM and others the first time they are used

- colloquial language 'getting adsorbed'

- imprecise language in places 'which was possibly estimated in crystal violet assay' - what does 'possibly estimated' mean?

- suggest reconsidering use of the word 'futuristic' (re: enzybiotics)

Reviewer #2 (Comments for the Author):

This paper focuses on the characterisation of phage PG14 that infects multidrug resistant *Klebsiella pneumoniae* and has potential for phage therapy. This subject area is vitally important from a global health perspective as carbapenem-resistant *K. pneumoniae* is an increasing clinical threat with few antimicrobial treatments available, phages could be increasingly important for therapy. This is a thorough and well written account of the isolation of a phage and its ability to degrade biofilms for future therapeutic applications.

Specific points

Try not to switch between CRKP and *K. pneumoniae*, stick to CR *K. pneumoniae* if you want to make the difference clear, but it is confusing to have CRKP and CLSM in the text as they are unfamiliar acronyms.

In "Importance" Line 25: I am not sure the statement "Biofilms responsible for device-related infections are mostly caused by *K. pneumoniae*." Is true, it is not backed up by references in the introduction.

Ensure all statements have references, e.g. Lines 41-2 add reference for "In addition, to its alarming antibiotic-resistant nature, *K. pneumoniae* displays a high degree of virulence enabling it to invade and survive in the host."

Lines 63-4 "several reports of phages ..." but the sentences only has one reference, add more to back up "several".

The text in all figures is too small, make text on figures and graph axes bigger.

Fig 3 The legends are too small to read. 3.B the colours for the range displayed are too close to see any differences, either removed colours to make it easier to read or make the rainbow across 94-100 (rather than current 50-100)

Crystal violet does not look like a good measure of *K. pneumoniae* biofilm (CLSM looks convincing), move Fig. 5 A to supplementary to concentrate on the important results. Explain why CV might not have been reliable

Fig 6 write "CLSM" out in full at the start of the legend (and in the results) so the figure can be understood on its own.

Fig 7 change the graph axes from "log PFU/ml" 6, 7, 8 to 1×10^6 , 1×10^7 , 1×10^8 .

Line 292 change "futuristic" to "future"

Line 299 states "CV staining assay, TVC assay and confocal microscopic analysis have confirmed the ability of phage PG14 to significantly inhibit and disrupt the biofilm" the results show that CV showed no biofilm degradation (Fig 5 A). Change the text.

Line 301 change "Eleanor M. Townsend et al 2021" to "Townsend et al 2021"

Line 328 what is "CLSI"? Write out the acronym in full

Line 330 change "Luria Bertani broth (LB)" to "Lysogeny Broth (LB)"

Lines 339, 365 change "0.22u" to "0.22um"

Lines 347-8 change "Luria Bertani agar Luria Bertani agar plate" to "Lysogeny Broth agar plate"

Line 383 add "."

Change reference 18 to "Townsend, EM, Moat, J, & Jameson, E. 2020. CAUTI's Next Top Model-model dependent *Klebsiella* biofilm inhibition by bacteriophages and antimicrobials. *Biofilm*, 2, 100038."

Staff Comments:

Preparing Revision Guidelines

- Point-by-point responses to the issues raised by the reviewers in a file named "Response to Reviewers," NOT IN YOUR

COVER LETTER.

- Upload a compare copy of the manuscript (without figures) as a "Marked-Up Manuscript" file.
- Each figure must be uploaded as a separate file, and any multipanel figures must be assembled into one file.
- Manuscript: A .DOC version of the revised manuscript
- Figures: Editable, high-resolution, individual figure files are required at revision, TIFF or EPS files are preferred

Please return the manuscript within 60 days; if you cannot complete the modification within this time period, please contact me. If you do not wish to modify the manuscript and prefer to submit it to another journal, please notify me of your decision immediately so that the manuscript may be formally withdrawn from consideration by Microbiology Spectrum.

Reviewer comments:

Reviewer #1 (Comments for the Author):

Summary: The manuscript contains useful data and represents a thorough characterization of the phage. It also has a nicely detailed methods section. However, the presentation of results sections (figures, figure legends, results) needs work.

Main issues:

Results section headings are generally not descriptive of the text, especially first two - I suggest rewriting them to emphasize the specific results. In addition, some results sections (body text) lack detail (one is only one sentence long).

Reply: Changes in the result headings and section have been made as per the suggestions. Refer line 79, 88, 93, and 127 to 136.

Most figure legends need more detail about what was done. Preferably the authors should get help with figure legend and figure conventions, and with stats.

Reply: All figures and their legends are modified as per the suggestion.

Figures are almost all blurry/hard to read, need work getting to publication quality.

Reply: Original figures with a good resolution are submitted separately.

Examples of issues with figures and figure legends:

- Aa / Ab / Bb naming in figs is confusing

Reply: Figure 1 legends are changed. Refer to lines 620 to 636.

Table 1

- EOP usually a quantitative value: what does each qualitative result correspond to? Should appear in legend. Would actually suggest a quantitative assessment if reporting as EOP

Reply: quantitative assessment of EOP is added to table 1.

Fig 1 A - difficult to see what's going on in this image. B - can't see plaque morphology due to angled blurry image

TEM - which phage is the one? many smaller heads too - looks like mixed sample

- temperature - should indicate how long treatment was

Reply: Fig 1 **a** is a field emission scanning electron microscopic image showing phage PG14 particles adsorbed on the cell surface of *K. pneumoniae* G14. The figure is modified by marking the area where the phages can be seen adsorbed on the bacterial cell surface. Figure 1**b** is modified to see the plaque morphology. In figure 1**c**, the TEM image is replaced with a new one with less background disturbance and marked to show the phage head and tail. Tiny structures in the background are probably salts/impurities commonly seen in TEM images.

Fig 4

- why mixed use of ## and * instead of */** / ***

Reply: In statistical analysis, levels of significance considered were from <0.001 to <0.05 ; hence

and * are both used

*p-value < 0.01

**p-value < 0.001 ,

***p-value < 0.0001

#p-value < 0.05

##p-value < 0.005

###p-value < 0.0005

Fig 5: legend specifically needs work (grammar, syntax, punctuation, capitalization)

Reply: Fig 5 legend is changed as per the suggestions. Refer to line 650 to 657.

fig 6 - should give more detail in legend about microscopy parameters used

Reply: Fig 6 legend is changed as per the suggestions. Microscopy parameters are added. Refer to line 658 to 665.

Fig 7 - check X axis labels, e.g. 'Control and log of PFU/ml of phage used for treatment..'

Reply: Fig 7 axis labels are changed as per the suggestions.

Fig 7 axis labels are inconsistent sizes, some blurry etc. Inconsistent labeling, ie. Fig7A is 'Time in Hour' and Fig 7B is 'Time of Treatment in Hours'

Reply: Fig 7 axis labels are changed as per the suggestions.

Fig 8 legend: run-on sentence, unclear

Reply: Fig 8 legend is changed as per the suggestion. Refer Line 682 to 688.

Results and discussion:

Host range: it would be nice to know why the strains chosen for host range were chosen (both in the text and in table 1).

Reply: Our laboratory works on alternative strategies to fight against multidrug-resistant organisms from the ESKAPE group of pathogens. We have an in-house collection of clinical isolates from ESKAPE group. Hence phage PG 14 isolated against *K. pneumoniae* G14 was tested for its host range against isolates of the ESKAPE group. Necessary changes are made in line 96 to 98 and 606 to 607.

Section: 'Antibiofilm activity of phage PG14'

- 'polysaccharide was its major component'. What kind of polysaccharide? how was this determined?

- very sparse section, need details on biochemical characterization mentioned

Reply: Modifications in the result section are done as per the suggestion. Refer line 128 to 136.

The main focus of our study was to isolate and characterize the phages against *Klebsiella*

pneumoniae and to study their antibiofilm activity. Therefore, only basic characterization of the biofilm matrix was carried out in the present study.

Discussion

- Very long run-on paragraphs
- check wording: 'to the suite of the basis for the assessment'

Reply: Paragraphs are modified as per the suggestion. Refer to lines 178 to 179, 193 to 196, 200 to 202, 205-206 and 217 to 221.

Writing:

The writing has several issues with sentence structure, punctuation, word choice, capitalization, inconsistencies, etc

Reply: We have thoroughly checked and revised the manuscript for all grammatical errors.

Examples:

- switch from S to +

Reply: Changed as per the suggestions.

- check comma usage throughout

Reply: Checked and changed as per the suggestions.

- check definition of opaque

Reply: Checked and changed as per the suggestions.

- define acronyms TEM, EOP, SEM and others the first time they are used

Reply: Acronyms are defined as per the suggestions.

- colloquial language 'getting adsorbed'

Reply: Checked and changed as per the suggestions.

- imprecise language in places 'which was possibly estimated in crystal violet assay' - what does 'possibly estimated' mean?

Reply: Changed as per the suggestion.

-suggest reconsidering use of the word 'futuristic' (re: enzybiotics)

Reply: Changed as per the suggestion.

Reviewer #2 (Comments for the Author):

This paper focuses on the characterization of phage PG14 that infects multidrug resistant *Klebsiella pneumoniae* and has potential for phage therapy. This subject area is vitally important from a global health perspective as carbapenem-resistant *K. pneumoniae* is an increasing clinical threat with few antimicrobial treatments available, phages could be increasingly important for therapy. This is a thorough and well written account of the isolation of a phage and its ability to

degrade biofilms for future therapeutic applications.

Specific points

Try not to switch between CRKP and *K. pneumoniae*, stick to CR *K. pneumoniae* if you want to make the difference clear, but it is confusing to have CRKP and CLSM in the text as they are unfamiliar acronyms.

Reply: Modified the statement as per the suggestion

In "Importance" Line 25: I am not sure the statement "Biofilms responsible for device-related infections are mostly caused by *K. pneumoniae*." Is true, it is not backed up by references in the introduction.

Reply: Modified the statement as per the suggestion and references are added. Refer lines 44 to 46

Ensure all statements have references, e.g. Lines 41-2 add reference for "In addition, to its alarming antibiotic-resistant nature, *K. pneumoniae* displays a high degree of virulence enabling it to invade and survive in the host."

Reply: Modified as per the suggestion. Refer line 40.

Lines 63-4 "several reports of phages ..." but the sentences only has one reference, add more to back up "several".

Reply: More references are added. Refer line 62.

The text in all figures is too small, make text on figures and graph axes bigger.

Reply: Modified as per the suggestion.

Fig 3 The legends are too small to read. 3.B the colours for the range displayed are too close to see any differences, either removed colours to make it easier to read or make the rainbow across 94-100 (rather than current 50-100)

Reply: Modified as per the suggestion.

Crystal violet does not look like a good measure of *K. pneumoniae* biofilm (CLSM looks convincing), move Fig. 5 A to supplementary to concentrate on the important results. Explain why CV might not have been reliable

Reply: Supplementary data is provided only for review purposes hence fig. 5a is kept in the manuscript. In crystal violet assay, live and dead biomass attached to the surface gets stained with crystal violet dye but only live cells are estimated when viable cell count of biomass is taken. In confocal laser scanning microscopy analysis, dead biomass (red fluorescence) is seen attached to the surface of wells in phage-treated samples which explains the difference observed in crystal violet assay and viable cell count.

Fig 6 write "CLSM" out in full at the start of the legend (and in the results) so the figure can be understood on its own.

Reply: Modified as per the suggestion.

Fig 7 change the graph axes from "log PFU/ml" 6, 7, 8 to 1×10^6 , 1×10^7 , 1×10^8 .

Reply: Modified as per the suggestion.

Line 292 change "futuristic" to "future"

Reply: Modified as per the suggestion.

Line 299 states "CV staining assay, TVC assay and confocal microscopic analysis have confirmed the ability of phage PG14 to significantly inhibit and disrupt the biofilm" the results show that CV showed no biofilm degradation (Fig 5 A). Change the text.

Reply: Modified as per the suggestion.

Line 301 change "Eleanor M. Townsend et al 2021" to "Townsend et al 2021"

Reply: Modified as per the suggestion.

Line 328 what is "CLSI"? Write out the acronym in full

Reply: Modified as per the suggestion.

Line 330 change "Luria Bertani broth (LB)" to "Lysogeny Broth (LB)"

Lines 339, 365 change "0.22u" to "0.22um"

Reply: Modified as per the suggestion.

Lines 347-8 change "Luria Bertani agar Luria Bertani agar plate" to "Lysogeny Broth agar plate"

Reply: Modified as per the suggestion.

Line 383 add "."

Reply: Modified as per the suggestion.

Change reference 18 to "Townsend, EM, Moat, J, & Jameson, E. 2020. CAUTI's Next Top Model-model dependent Klebsiella biofilm inhibition by bacteriophages and antimicrobials. *Biofilm*, 2, 100038."

Reply: Modified as per the suggestion.

October 19, 2022

Dr. Karishma Rajendra Pardesi
Savitribai Phule Pune University, Pune, Maharashtra, India
Department of Microbiology
Ganeshkhind road
Pune, Maharashtra 411007
India

Re: Spectrum01994-22R1 (Characterization of novel Klebsiella phage PG14 and its antibiofilm efficacy)

Dear Dr. Karishma Rajendra Pardesi:

Your manuscript has been accepted, and I am forwarding it to the ASM Journals Department for publication. You will be notified when your proofs are ready to be viewed. Regarding publication costs, the Spectrum journal staff will work with you to find a suitable solution. The publication waiver policy is available here for your reference: <https://journals.asm.org/waivers>

Sincerely,

Daria Van Tyne
Editor, Microbiology Spectrum
